# Microstructure and Mechanical Performance of Resistance Spot Welded Martensitic Advanced High Strength Steel

**Yunzhao Li [1,2], Huaping Tang [1] and Ruilin Lai [3,\*]**

1 State Key Laboratory of High Performance Complex Manufacturing, Central South University, Changsha 410083, China; liyunzhao207@163.com (Y.L.); huapingt-csu@163.com (H.T.)
2 Hunan Railway Professional Technology College, Zhuzhou 412001, China
3 State Key Lab of Powder Metallurgy, Powder Metallurgy Research Institute, Central South University, Changsha 410083, China
\* Correspondence: 133701033@csu.edu.cn; Tel.: +86-181-6365-0762

**Abstract:** Resistance spot welded 1.2 mm (t)-thick 1400 MPa martensitic steel (MS1400) samples are fabricated and their microstructure, mechanical properties are investigated thoroughly. The mechanical performance and failure modes exhibit a strong dependence on weld-nugget size. The pull-out failure mode for MS1400 steel resistance spot welds does not follow the conventional weld-nugget size recommendation criteria of $4t^{0.5}$. Significant softening was observed due to dual phase microstructure of ferrite and martensite in the inter-critical heat affected zone (HAZ) and tempered martensite (TM) structure in sub-critical HAZ. However, the upper-critical HAZ exhibits obvious higher hardness than the nugget zone (NZ). In addition, the mechanical properties show that the cross-tension strength (CTS) is about one quarter of the tension-shear strength (TSS) of MS1400 weld joints, whilst the absorbed energy of cross-tension and tension-shear are almost identical.

**Keywords:** resistance spot welding; martensitic steel; mechanical performance; microstructure; failure mode

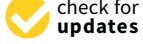



## 1. Introduction

With the development of science and technology and improvements in the standard of living, the safety, energy saving and environmental protection of automobiles has increasingly become the research focus of interest. Thus, advanced high strength steels (AHSS), due to their excellent strength and formability, have accelerated the integration of AHSS into the automotive architecture to reduce weight, improve fuel efficiency and increase the crashworthiness of vehicles through enhanced mechanical properties [1–5].

The welding methods used by the automotive manufacturing industry include resistance spot welding, laser welding, and arc welding [6,7]. High heat input during the laser welding and arc welding process causes the heat affected zone (HAZ) to soften in AHSS, due to the tempered martensite in the base metal, which in turn affects the mechanical properties [8–11]. With excellent adaptability, sound quality assurance and high efficiency, resistance spot welding (RSW) is the most predominant welding technique adopted for joining AHSS in automotive applications [12–14]. Typically, auto-body assembly needs 3000 to 5000 spots of welding, so the quality and mechanical performance of the resistance spot welds are very significant to the durability and reliability of the vehicles [15,16]. Generally, there are three critical factors for the quality evaluation of resistance spot welds, including weld-nugget size, weld mechanical performance and failure mode. Weld-nugget size is the most important parameter determining weld mechanical performance and failure mode. The tension-shear and cross-tension test are the most widely used approach to evaluate spot weld mechanical behaviours. Besides, interfacial and pull-out failure mode are the two major types of spot weld failures to qualitatively estimate the weld quality. Many researchers have demonstrated that the AHSS samples prepared by resistance spot welds

are highly susceptible to failure during the interfacial mode, thereby resulting in a lowered mechanical performance [14,15,17–20]. The weld failure mode and mechanical properties are also highly dependent on the microstructure of NZ/HAZ/base metal. Pouranvari et al. [12,21,22], when studying the metallurgical response of martensitic AHSS joints during RSW, found that a significant softening was observed in the heat affected zone (HAZ) due to allotriomorphic ferrite formation in the inter-critical HAZ and the tempering of martensite in sub-critical HAZ. Meanwhile, Rezayat et al. [23] also found such a softening of HAZ facilitates the enhancement of the ultimate load and global extension in high strength grades steels in martensitic AHSS.

In spite of the aforementioned results, there is still a lack of a deeper and more systematic investigation into the process–microstructure–performance relationships in resistance spot welding of martensitic advanced high strength steel. Therefore, the aim of this study is to investigate the weld behaviour of MS1400 RSW under tension-shear and cross-tension tests. The microstructures and mechanical properties, including cross-tension strength (CTS) and tension-shear strength (TSS) and absorbed energy, regarding the MS1400 steel resistance spot welds were also analysed. The fracture mode was revealed by fractography analyses.

## 2. Materials and Methods

An uncoated cold rolled MS1400 martensitic AHSS (Docol 1400 M) sheet 1.2 mm thick was concerned as the base-metal (BM). The chemical composition and mechanical properties of the MS1400 steel are shown in Table 1. The microstructure of MS1400 is basically tempered martensite [1]. Note that the chemical composition and mechanical properties shown in Table 1 are collected from the manufacturers.

**Table 1.** The chemical composition and mechanical properties of the MS1400.

| Chemical Composition (wt. %) | | | | | | Mechanical Properties | | |
|---|---|---|---|---|---|---|---|---|
| C | Si | Mn | P | S | Al | Yield Strength (MPa) | Tensile Strength (MPa) | Elongation (%) |
| 0.17 | 0.42 | 1.66 | 0.009 | 0.002 | 0.03 | 1338 | 15,184 | 4 |

The 440 kVA medium frequency inverter DC seat resistance spot welding machine is used for spot welding. The working frequency is 50 Hz, and the welding are controlled by PLC. The electrodes were dome-type, and the diameter of electrode surface is 6 mm. The quasi-static cross-tension and tensile-shear test samples are prepared according to the ANSI/AWS/SAE/D8.9-2012 standard [24]. As the tension shear specimen is asymmetric, two gaskets with the same thickness are added to the clamping part of the specimen to ensure alignment and reduce plate bending and nugget rotation (cf. Figure 1). The cross tensile and tensile shear tests were carried out on a universal material testing machine at the speed of 2 mm/min. At least 5 individual samples are conducted for the cross tensile and tensile shear tests to guarantee the reliability.

To examine the quality of MS1400 resistance spot welding steel, the microstructure, and mechanical properties for a cross section of the sample are analysed by optical micrograph and micro-hardness, respectively. The polished weld MS1400 steel joints were etched by nital (i.e., 4 vol.% $HNO_3$ + 96 vol.% ethanol). The microstructure of the weld zone is also observed by a scanning electron microscope (SEM). The microhardness of spot welding is tested according to the procedure specified in ANSI/AWS/SAE/D8.9-2012 [24]. All the measurements are the average of three points measured diagonally in the thickness direction of the weld on the cross section of the specimen, starting from the base metal of the first plate, passing through the nugget, and then entering the base metal of the second plate. The microhardness of metallographic specimens corroded by 4% nital is measured by Vickers indenter under a 500 g load and 10 s hold time. The error bar represents a standard deviation of the hardness results. The fracture surface is observed by SEM under the mode of secondary electron (SE).

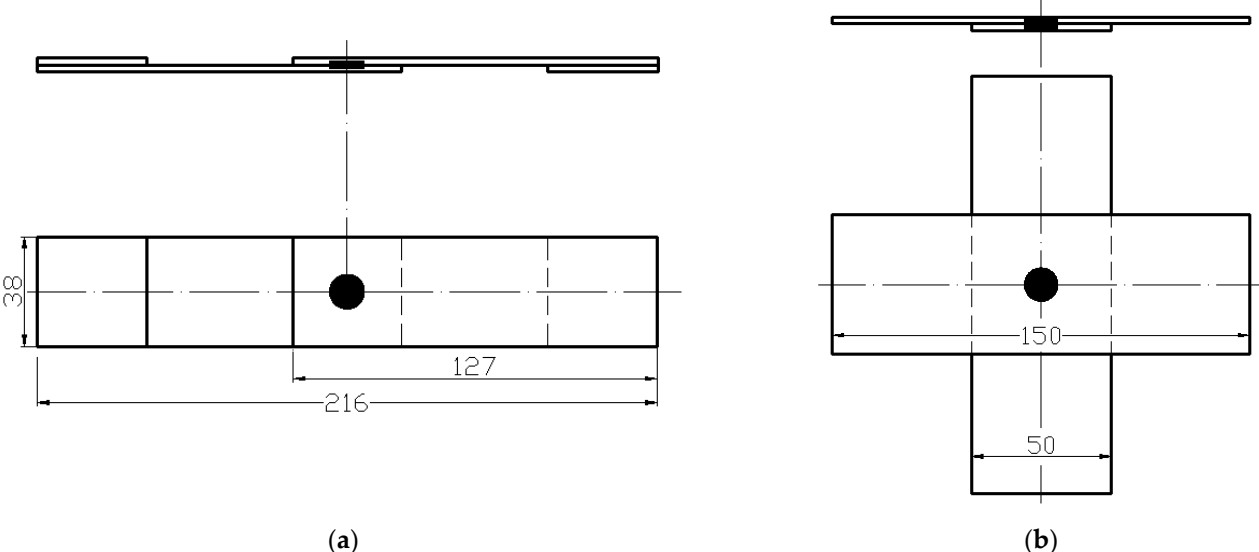

(**a**)                                                        (**b**)

**Figure 1.** Schematics of resistance welded (**a**) tension-shear, and (**b**) cross-tension specimens (dimensions in mm).

## 3. Results and Discussion

### 3.1. Effect of Welding Parameters on Mechanical Properties of Welded MS1400

The welding process should be in accordance with ANSI/AWS/SAE/D8.9-2012 [24]. The electrode force values is fixed at 4000 N, the welding time and welding current are varied to determine their roles in the mechanical properties of welded MS1400. The failure modes of the weld joints of TSS and CTS for MS1400 steel are shown in Figures 2 and 3, respectively. The hollow, semi-solid, and solid round symbols represent the interfacial weld fracture, the mixed interfacial and weld pull-out failure modes, and weld pull-out, respectively.

It can be observed from Figure 2 that TSS exhibits a parabolic characteristic with the increase in welding current and welding time. A similar phenomenon is also observed for the welded CTS joints, as shown in Figure 3. Both the TSS and CTS MS1400 reaches its respective maximum value (21 and 6 kN) concurrently under welding current 8 KA and welding time of 440 ms. However, when the welding current further increases to 9 kA, the TSS and CTS decline rapidly due to the occurring expulsion.

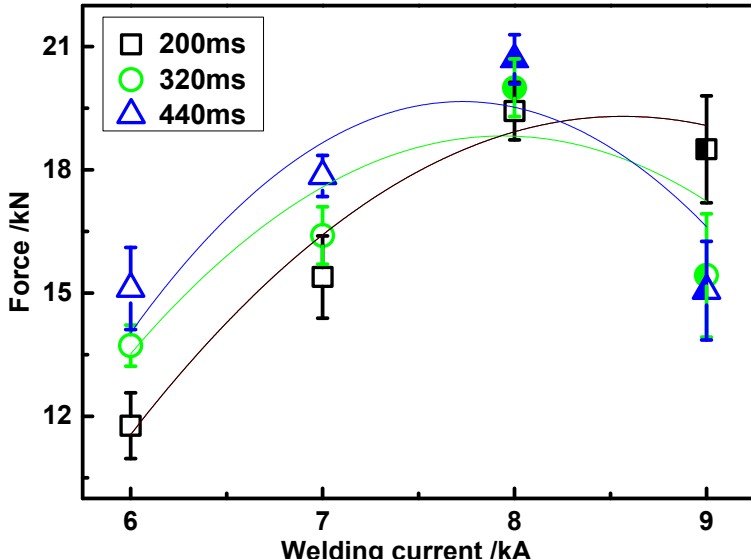

**Figure 2.** TSS and failure modes of resistance spot welded 1.2 mm thick MS1400 steel.

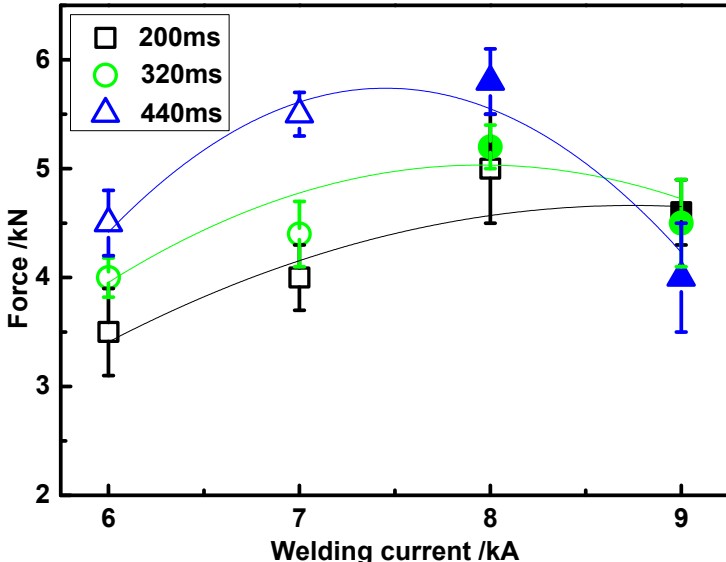

**Figure 3.** CTS and failure modes of resistance spot welded 1.2 mm thick MS1400 steel.

Weld-nugget size is one of the most important parameters to determine the mechanical properties of the welded joints. It should be mentioned that some welding parameters, i.e., welding current and welding time, have significant effects on weld-nugget size (cf. Figure 4). To be specific, the weld-nugget size increases with the increase of welding current and welding time. However, when the welding current increases to be higher than 9 kA, the weld-nugget size decreases owing to occurring expulsion.

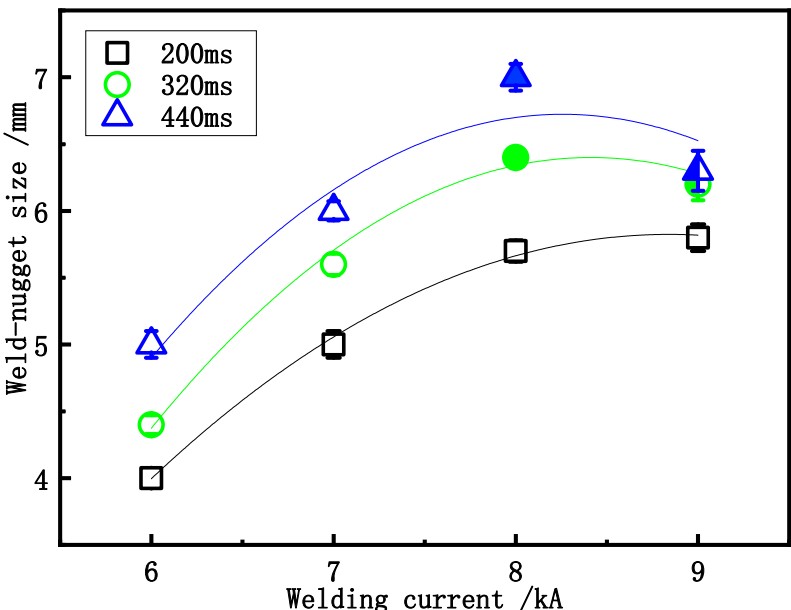

**Figure 4.** Effects of welding current and welding time on the weld nugget size.

The weld lobe, which is the window of applicable weld parameters, is mainly determined by the minimum weld-nugget size and by the expulsion limit. From Figure 5, all the failure modes show perspicuous interfacial fracture when the welding currents are equal to or less than 7 kA. Pull-out failure modes by tension-shear and cross-tension tests occur above an applied current of 8 kA, with welding times higher than 300 ms. The expulsion limit is exceeded due to welding with excessive heat input when the welding current reaches 9 kA. The fracture modes of cross-tension tested welds are pull-out, whilst the

interfacial fracture mode dominates at tension-shear tested welds due to expulsion caused by severe overheating [17]. In Figure 5, such a narrow acceptable welding current range is observed at about 1 kA, which is much narrower than other type of AHSS [1,25,26].

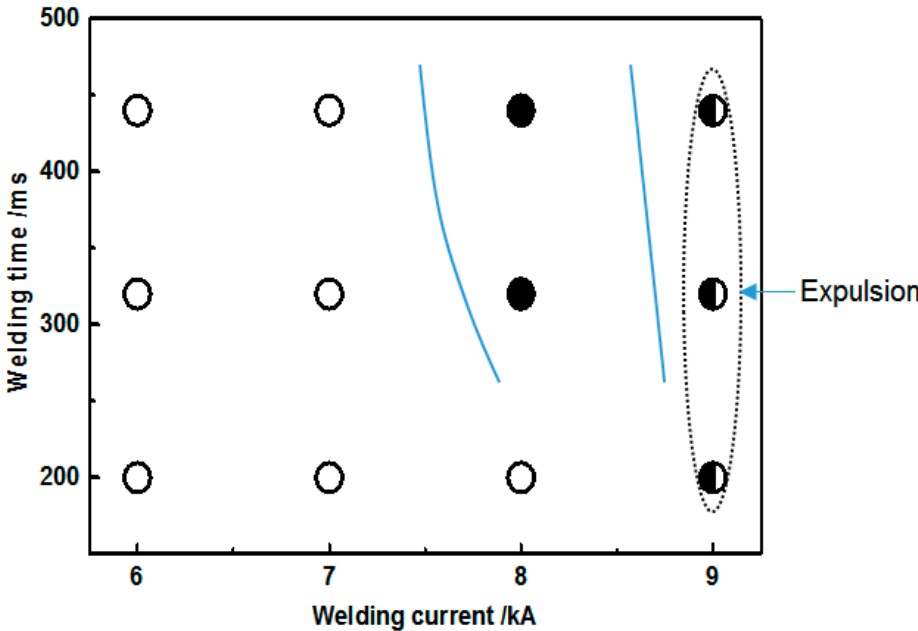

**Figure 5.** Welding lobes for resistance welding of 1.2 mm thick MS1400 steel.

The MS1400 weld still belongs to the interfacial fracture mode under tension-shear and cross-tension load, even if the weld-nugget size is larger than the conventional recommended weld-nugget size of $4t^{0.5}$ (cf. Figures 4 and 5). Note that the minimum weld-nugget size required to ensure the failure mode in tension-shear and cross-tension tests is larger than 6.4 mm, based on Figure 4. Therefore, the traditional weld-nugget size criterion [24] is not enough to ensure that the pull-out failure mode of MS1400 weld-nugget takes place. According to Japanese JIS Z 3140 [27] and German DVS 2923 [28] standards, the acceptable weld-nugget size is specified with $5t^{0.5}$. For practical purposes, the equation for the critical weld-nugget size ($D_C$) for 1.2 mm thickness MS1400 AHSS can be derived as follows:

$$D_C = 6t^{0.5} \approx 6.6 \text{ mm} \tag{1}$$

### 3.2. Microstructure of Welded MS1400

The hardness profiles of MS1400 RSW, along with a typical macrostructure, are conducted under 4 kN welding force, 8 kA welding current and 320 ms welding time, with the results shown in Figure 6. Based on these, three typical regions, BM, HAZ and NZ, are distinguished clearly. The HAZ can be further sub-divided into upper-critical HAZ (UCHAZ), inter-critical HAZ (ICHAZ) and sub-critical HAZ (SCHAZ).

In terms of the microstructure of MS1400 before welding, fine grains of martensite are observed (cf. Figure 7a) and corresponding average hardness can reach 480 HV.

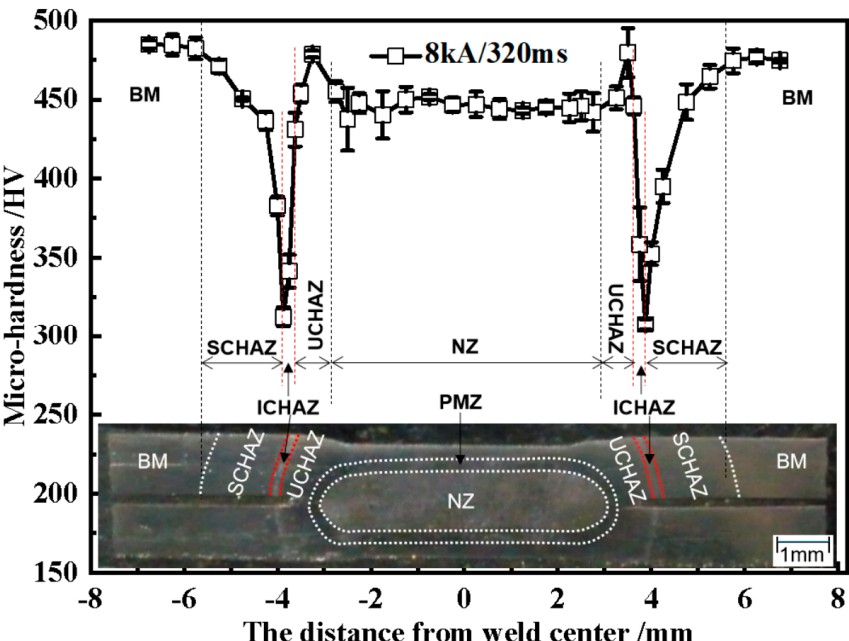

**Figure 6.** Hardness profiles and corresponding macrostructure of MS1400 weld joint.

The NZ is mainly lath martensite (cf. Figure 7b) with an average hardness of 450 HV. Compared with BM with fine martensite grains (cf. Figure 7a), the formation of coarse columnar lath martensite in NZ should be responsible for the decrease in hardness. Such a formation of martensite in NZ is attributed to the high cooling rate inherent in the resistance spot welding process, which is due to the existence of a water-cooled copper electrode and its quenching effect, as well as the short welding cycle. The phase transformation sequence of the UCHAZ is as follows:

$$M \xrightarrow{\text{Heating}} L \xrightarrow{\text{Cooling}} \gamma \xrightarrow{\text{Cooling}} M \tag{2}$$

Compared to the NZ, a large microstructural gradient in the HAZ containing UCHAZ, ICHAZ, and SCHAZ are strikingly apparent (cf. Figure 7c). A magnified view of the SCHAZ microstructure is shown in Figure 7d. The material in this zone experiences a peak temperature which is below the $A_{c1}$ line, thus the existing martensite in the BM of the MS1400 undergoes a tempering process in SCHAZ. During tempering, the martensite is severely decomposed by carbon atoms diffusion from the carbon supersaturated martensite phase, thus various shapes of carbides such as cementite ($Fe_3C$) are formed. The broken lath morphology and the presence of a high density of precipitated carbides uniformly distributed throughout the tempered martensite (TM) structure in SCHAZ can be seen in Figure 7d. The decomposition of martensite plays an important role on the significant softening of the MS1400 welds in the SCHAZ. The phase transformation sequence of the SCHAZ is as follows:

$$M \xrightarrow{\text{Heating}} TM \xrightarrow{\text{Cooling}} TM \tag{3}$$

Figure 7e displays the microstructure of the ICHAZ, exhibiting dual-phase microstructures of ferrite and martensite. The maximum temperature of ICHAZ is between $A_{c1}$ and $A_{c3}$. This causes the partial transformation of the base metal into austenite and ferrite. Note that all the austenite are transformed into martensite with a rapid cooling rate. Therefore, two phases of martensite and ferrite are mixed in the ICHAZ region, as clearly shown in Figure 7e. Owing to the softened microstructure, the hardness of HAZ is lower than that of BM and NZ composed fully of martensite. The phase transformation sequence of the ICHAZ is as follows:

$$M \xrightarrow{\text{Heating}} \gamma + \alpha_F \xrightarrow{\text{Cooling}} M + \alpha_F \tag{4}$$

Figure 7f shows the microstructure of the UCHAZ. The temperature of UCHAZ during RSW is above $A_{c3}$. In this temperature range, the as-received steel is fully austenitized. Owing to carbon-rich austenite and the high cooling rate, an almost fully martensitic is obtained. It can be seen from Figure 7f that the UCHAZ of spot welded MS1400 exhibits a fully martensitic structure. It should be noted that the hardness in the UCHAZ is slightly higher than that of NZ, which is attributed to the finer grain size of the UCHAZ in comparison with the NZ (cf. coarsened columnar grains shown in Figure 6). The phase transformation sequence of the UCHAZ is as follows:

$$M \xrightarrow{\text{Heating}} \gamma \xrightarrow{\text{Cooling}} M \tag{5}$$

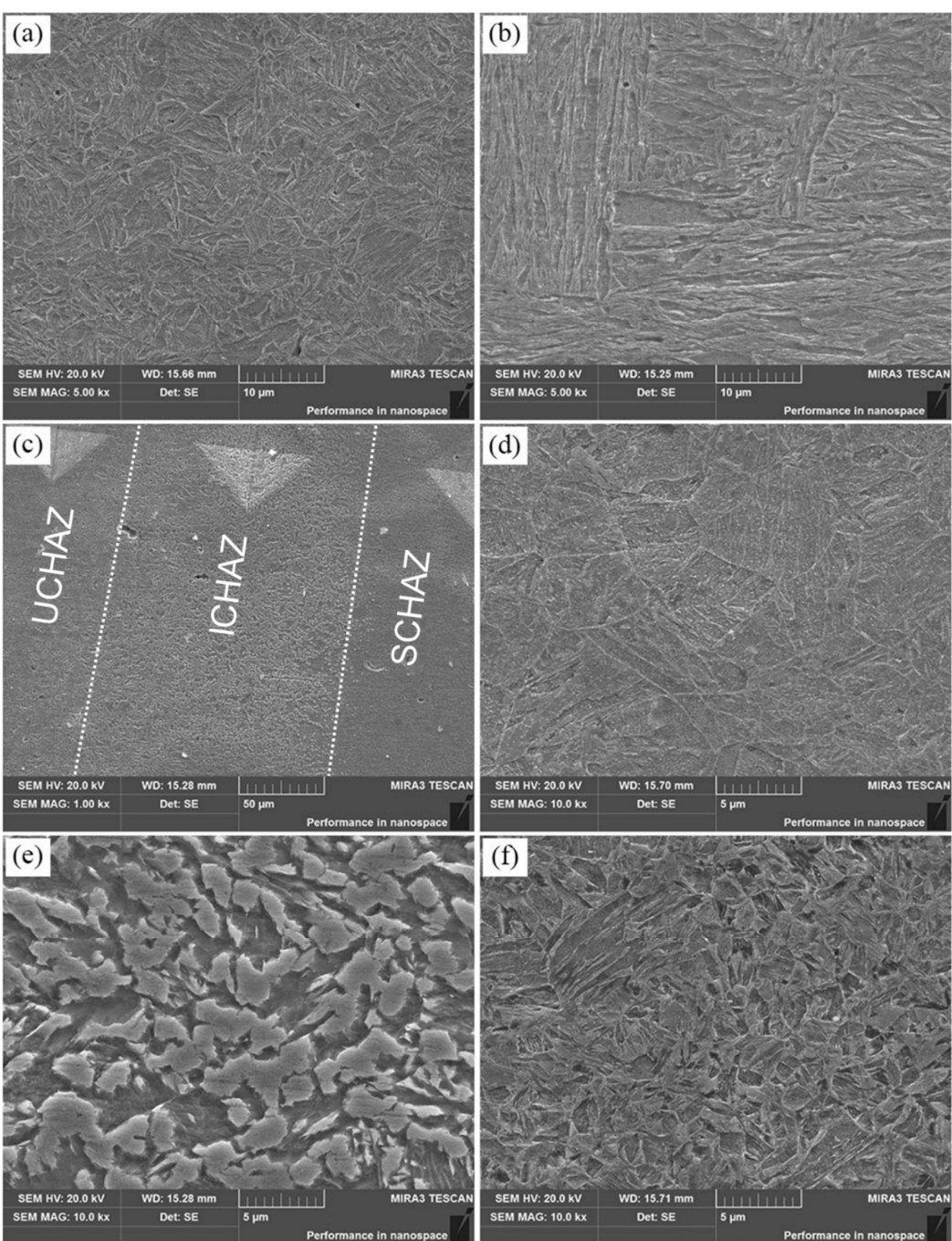

**Figure 7.** SEM micrographs showing microstructure of (**a**) BM, (**b**) NZ, (**c**) HAZ, (**d**) SCHAZ, (**e**) ICHAZ, (**f**) UCHAZ for RSW on MS1400 steel.

### 3.3. Absorb Energies and Fracture Behavior under Cross-Tension and Tension-Shear Test

The load displacement curves associated with the cross-tension and tension-shear test are shown in Figure 8. Although the load for CTS (5 kN) is distinctly lower than that for TSS (19.9 kN), the displacement for the cross-tension test (10.4 mm) is significantly higher than that of the tension-shear test (3.0 mm). In addition, the absorbed energy of cross-tension and tension-shear is shown in Figure 8. To completely explore the performance of the spot welds of MS1400, the absorbed energy of cross-tension and tension-shear were measured (see the shadow area in Figure 8). Although the CTS is nearly one quarter of TSS, the absorbed energy of the cross-tension test is near identical in value to that of the tension-shear test.

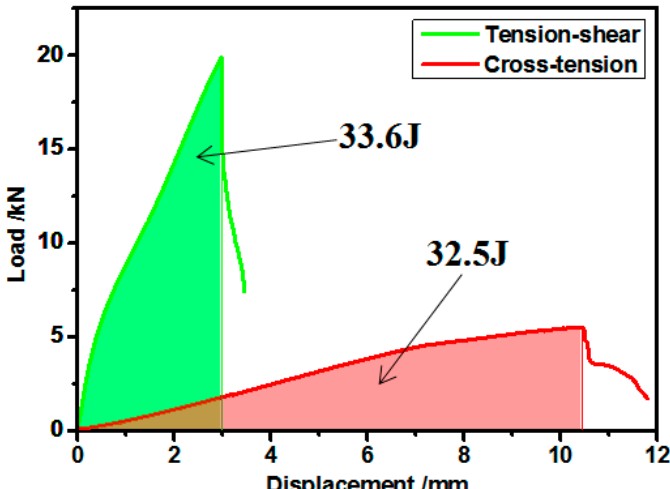

**Figure 8.** The load-displacement of the resistance welded MS1400 steel joints produced by 4 kN welding force and 8 kA welding current and 320 ms welding time.

The pull-out failure mode of the cross-tension and tension-shear test are investigated by SEM. The elongated (shear-type) dimples in Figure 9e indicate that the fracture mechanism of tension-shear is ductile, and the failure is introduced by shear stress, which is consistent with the finding by Pouranvari et al. [29]. Figure 9b shows a typical fracture surface during the cross-tension tests, and Figure 9d shows the SEM image of the fractured surface of cross-tension. It can be clearly observed that the failure appears to be initiated at the nugget circumference, and finally, the shear fracture penetrates the sheet thickness direction. Fracture surfaces at the initiated surface exhibit brittle cleavage, as well as small areas of ductile fracture, as illustrated in the SEM magnified image of Figure 9f. This is in accordance with common semi-brittle fractures in hard martensitic microstructures that can be found in the weld nugget of relatively high alloyed DP or TRIP steels [30]. The elongated (shear-type) dimples in Figure 9g indicate that the fracture end of the cross-tension is ductile and has the shape of parabolas pointing in the loading direction. It reveals that the driving force for pull-out failure mode in the cross-tension test is a combination of tensile stress and shear stress.

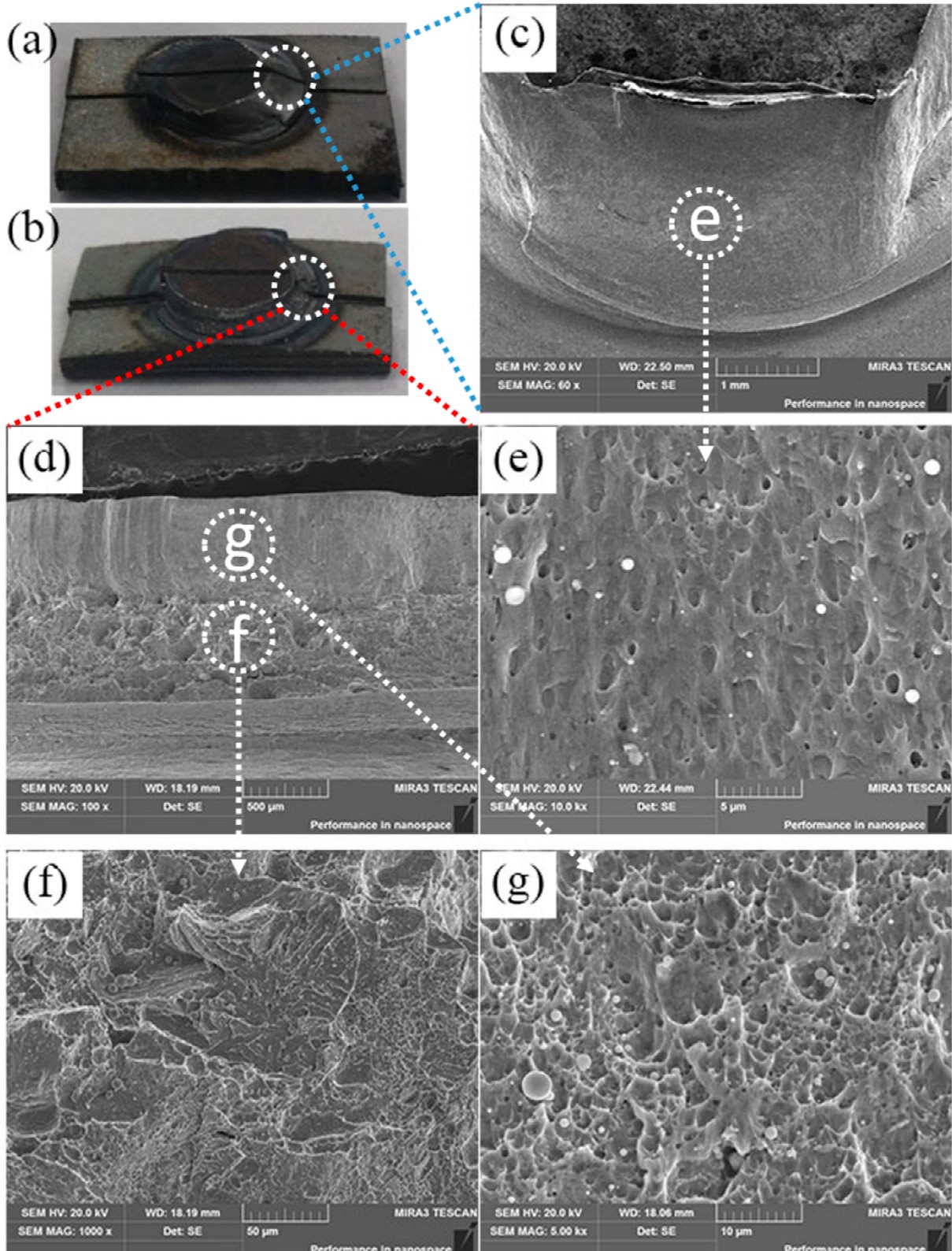

**Figure 9.** Typical fracture: (**a**) tension-shear and (**b**) cross-tension test, SEM images shows the fractographs of the fractured surface: (**c**) tension-shear and (**d**) cross-tension and the magnified images of the selected zone with marked by (**e**–**g**).

## 4. Conclusions

The microstructure, mechanical performance and post-mortem fracture characteristics of resistance spot welds of MS1400 are investigated. The main conclusions are summarized as follows:

1. Tension-shear strength (TSS), cross-tension strength (CTS) and weld-nugget size are significantly affected by welding current and welding time. The existing industrial weld-nugget size criteria are not sufficient to ensure the pull-out failure mode during the tensile-shear and cross-tension test of the MS1400 resistance spot welds. The proposed can predict the failure mode with good accuracy. With good accuracy, a practical model for the critical weld-nugget size ($D_C = 6t^{0.5}$ ) for 1.2 mm thickness MS1400 AHSS is derived.

2. The formation of coarsened columnar lath martensite in the NZ lead to a decrease in hardness of the NZ compared to that of BM with fine martensitic grain. The HAZ softening is attributed to the tempered martensite (TM) structure in SCHAZ and dual phase microstructure of ferrite and martensite in the ICHAZ region. The UCHAZ fully exhibits a martensitic microstructure.

3. The CTS is nearly one quarter of TSS, the absorbed energy of the cross-tension test is near identical in value to that of the tension-shear test. The failure mechanism of tension-shear, which fails via pull-out mode, was introduced by shear stress. However, the driving force for pull-out failure mode in the cross-tension test is a combination of tensile stress and shear stress.

**Author Contributions:** R.L. was the principal investigator of the research. Y.L. performed the experiment and analyzed the data with the assistance of R.L. and H.T. provided the devices used for the experiment. Y.L., H.T., and R.L. wrote the manuscript with feedbacks from all the authors. All authors have read and agreed to the published version of the manuscript.

**Funding:** This research was supported by National Key Research and Development Program of China (SQ2019YFA070172).

**Institutional Review Board Statement:** Not applicable.

**Informed Consent Statement:** Not applicable.

**Data Availability Statement:** Data is contained within the article.

**Acknowledgments:** The authors gratefully acknowledge Pei-Chung Wang, who work at the Manufacturing Systems Research Lab of General Motors Global Research and Development.

**Conflicts of Interest:** The authors declare no conflict of interest.

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
