# Peer review of "Microstructure and Mechanical Performance of Resistance Spot Welded Martensitic Advanced High Strength Steel"

_processes, doi:10.3390/pr9061021_

Round 1

Reviewer 1 Report

In this paper, the resistance spot welding characteristics of martensitic AHSS were evaluated through microstructural and mechanical analysis methods. The content of the paper did not have any major contradictions, and the analysis results are considered to be suitable in supporting the author's argument. I think this paper will be suitable for publication in this journal if the following is revised.

  • There are not enough recent literature surveys studies on the resistance spot welding characteristics of martensitic AHSS. Of course, it is understood that limited research has been conducted so far, but additional surveys on the latest research trends are needed.
  • In Figures 2, 3, and 5, the symbols of the fracture mode are marked separately (hollow, semi-solid, and solid round symbols). However, in Fig. 4, the symbols are not used differently depending on the fracture mode, which may cause confusion among readers. Therefore, it is recommended that the symbol in Fig. 4 be represented according to the fracture mode as in other figures.

Typo

  • Line 113: 9ka  --> 9kA
  • Line 119: Fig 5 -->Figure 5

Author Response

Dear Ms. Zora Zhou and reviewer:

Thank you very much for your kind suggestions on the revision of our manuscript entitled Microstructure and Mechanical Performance of Resistance Spot Welded Martensitic Advanced High Strength Steel (Manuscript ID: processes-1236559) submitted for publication in Processes. Meanwhile, we appreciate very much for the kindness of the reviewers, who gave us so many valuable comments and sound suggestions to the improvement of our manuscript. The revisions have been carefully carried out according to the reviewers’ and your comments. The revised parts are marked up using the “Track Changes” function in the revised manuscript. Reviewer’s comments and the corresponding revisions are listed as follows:‬‬

Reviewer: 1

  1. There are not enough recent literature surveys studies on the resistance spot welding characteristics of martensitic AHSS. Of course, it is understood that limited research has been conducted so far, but additional surveys on the latest research trends are needed.
  • Response: Thanks for your constructive suggestions, we have added necessary recent literature surveys studiesin the Introduction.(line 33) 
  1. 2. In Figures 2, 3, and 5, the symbols of the fracture mode are marked separately (hollow, semi-solid, and solid round symbols). However, in Fig. 4, the symbols are not used differently depending on the fracture mode, which may cause confusion among readers. Therefore, it is recommended that the symbol in Fig. 4 be represented according to the fracture mode as in other figures.
  • Response:Thanks for your constructive suggestions, we have corrected it in the modified manuscript.(line 119)
  1. 3. Typo

Line 113: 9ka  --> 9kA

Line 119: Fig 5 -->Figure 5

  • Response: It’s our typo mistake. We have corrected it in the modified manuscript . Weare glad for your careful review.

In summary, the whole manuscript has been revised carefully according to the reviewers’ comments and suggestions. We would be very grateful if the revised manuscript could be finally accepted for publication in Processes. Thanks for your consideration in advance and we are looking forward to hearing from you at your earliest convenience.

With best regards!

Yours sincerely,

Ruilin Lai, PhD

Central South University, China

Reviewer 2 Report

Dear Authors,

I have read paper "Microstructure and Mechanical Performance of Resistance Spot Welded Martensitic Advanced High Strength Steel".

Paper fulfills the aims and scope of Processes. After some improvements, it could be considered for publishing. My questions and suggestions are listed below.

General remarks:

  • The references should be numbered.
  • Your references are unacceptable. None have been published in last three years. The newest has been published in 2016. It is imposible to show relevant scientific background without latest articles. The science made big step forward last years. The usage of martensitic steels is increasing annually. You have to rebulit your referneces - please add many of latest articles (last three years).
  • Please check the abstract - different styles appeared.
  • TSS and CTS are not described in the abstract. Please show the full names.

Introduction:

  • This section should be seriously modified by newly published information. You should mark the problems connected with joining the investigated steels. The processes of joining should be also describes the same, as usage.
  • You should underline, why RSW was chosen for investigations. Why, other methods has not been considered - please mark the problems with base material during other joining processes.
  • The novelty should be marked.

Materials and Methods:

  • Line 52 "The material used in the researchwas a kind of 1400 MPa grade of AHSS" - this type of description is unacceptable. You have to straightly mark the grade of used steel.
  • Table 1 - the source of presented values is unknown. Have you analysed this content? Or it was taken from standard or manufacturer data? Is has to be marked in the text.
  • Please show the mechanical properties of used steel.
  • How many specimens have you performed? It is not clear.

Results and Discussion:

  • How you assessed the weldability? There are analytical methods to assess the weldability (e.g., evaluation of carbon equivalent). The weldability could be assessed by standard weldability tests, which is not mentioned in your paper. The most comonly used methods are Tekken and CTS tests in accordance with ISO 17642-2:2005 (e.g., 10.3390/app10051823 . Moreover, the ISO 15614-1:2017 standard, which allows to qualification the welding procedure is used for weldability evaluation (e.g., 10.3390/ma13235535). Additionally tests could be performed. However, I cannot find any information about performed weldability tests. You have measures some parameters, calculated some values. It is proper to name this as "weldability"? Please support with references/standards.

  • The discussion should be extended. Please compare your results with literature more deeply. You should mark the advantages of your investigations.

Conclusions:

  • Whis part is the strongest. Conclusions are supported with results from experiments.

Author Response

Dear Ms. Zora Zhou and reviewer:

Thank you very much for your kind suggestions on the revision of our manuscript entitled Microstructure and Mechanical Performance of Resistance Spot Welded Martensitic Advanced High Strength Steel (Manuscript ID: processes-1236559) submitted for publication in Processes. Meanwhile, we appreciate very much for the kindness of the reviewers, who gave us so many valuable comments and sound suggestions to the improvement of our manuscript. The revisions have been carefully carried out according to the reviewers’ and your comments. The revised parts are marked up using the “Track Changes” function in the revised manuscript. Reviewer’s comments and the corresponding revisions are listed as follows:‬‬

Reviewer: 2

  1. The references should be numbered.
  • Response: Thanks for your soundadvice, we have numbered the references in the modified manuscript.

  1. Your references are unacceptable. None have been published in last three years. The newest has been published in 2016. It is imposible to show relevant scientific background without latest articles. The science made big step forward last years. The usage of martensitic steels is increasing annually. You have to rebulit your referneces - please add many of latest articles (last three years).
  • Response: Thanks for your constructive Based on your comments, we have added many of latest references in the revised manuscript. 
  1. Please check the abstract - different styles appeared.
  • Response: Thanks for your beneficial We have revised the abstract in the modified manuscript.
  1. TSS and CTS are not described in the abstract. Please show the full names.
  • Response: Thanks for your careful review, we have added the full namesin the modified manuscript.
  1. This section should be seriously modified by newly published information. You should mark the problems connected with joining the investigated steels. The processes of joining should be also describes the same, as usage.
  • Response: Thanks for your constructive suggestions. We have carefully modified by newly published information and added necessary recent literature surveys studiesin the Introduction.
  1. You should underline, why RSW was chosen for investigations. Why, other methods has not been considered - please mark the problems with base material during other joining processes.
  • Response: Thanks for your constructive suggestions. The welding methods used by the automotive manufacturing industry include resistance spot welding, laser welding, and arc welding [6, 7]. High heat input during laser welding and arc welding process, causes heat affected zone (HAZ) to soften in AHSS due to martensite in the base metal is tempered, which in turn affects the mechanical properties [8-11]. With excellent adaptability, sound quality assurance and high efficiency, resistance spot welding (RSW) is the most predominant welding technique adopted for joining AHSS in automotive applications. Correspondinghave added in the Introduction into the revised manuscript.(line 33)
  1. The novelty should be marked.
  • Response:Thanks for your constructive suggestions.The novelty have added in the revised manuscript.(line 60)
  1. Line 52 "The material used in the researchwas a kind of 1400 MPa grade of AHSS" - this type of description is unacceptable. You have to straightly mark the grade of used steel.
  • Response: Based on your advice, we have modified this type of description in the revised manuscript. Thanks for your constructive suggestions!(line 68)
  1. Table 1 - the source of presented values is unknown. Have you analysed this content? Or it was taken from standard or manufacturer data? Is has to be marked in the text.
  • Response: Thanks for your constructive suggestions. The source of presented values was taken from the manufacturer data and we have marked in the revised manuscript.(line 72)
  1. Please show the mechanical properties of used steel.
  • Response: Thanks for your constructive suggestions.We have added the mechanical properties of used steel in the revised manuscript.(line 73)
  1. How many specimens have you performed? It is not clear.
  • Response: Thanks for your constructive suggestions.At least 5 individual samples are conducted for the cross tensile and tensile shear tests, to guarantee the reliability. Corresponding descriptions were also added in the revised manuscript. (line 83)
  1. How you assessed the weldability? There are analytical methods to assess the weldability (e.g., evaluation of carbon equivalent). The weldability could be assessed by standard weldability tests, which is not mentioned in your paper. The most comonly used methods are Tekken and CTS tests in accordance with ISO 17642-2:2005 (e.g., 10.3390/app10051823 . Moreover, the ISO 15614-1:2017 standard, which allows to qualification the welding procedure is used for weldability evaluation (e.g., 10.3390/ma13235535). Additionally tests could be performed. However, I cannot find any information about performed weldability tests. You have measures some parameters, calculated some values. It is proper to name this as "weldability"? Please support with references/standards.
  • Response: Thanks for your positive comments.Based on your advice, we have further analyzed our data and we found it is not proper to name this as “weldability”. Thus, we have modified this part in the revised manuscript.
  1. The discussion should be extended. Please compare your results with literature more deeply. You should mark the advantages of your investigations.
  • Response: Thanks for your positive comments.Based on your advice, we have further extended the discussion in the revised manuscript.
  1. Whis part is the strongest. Conclusions are supported with results from experiments.
  • Response: Thanks for your constructive suggestions, which help us improve our manuscript greatly. We have revised the conclusionsin the revised manuscript (line 234).

In summary, the whole manuscript has been revised carefully according to the reviewers’ comments and suggestions. We would be very grateful if the revised manuscript could be finally accepted for publication in Processes. Thanks for your consideration in advance and we are looking forward to hearing from you at your earliest convenience.

With best regards!

Yours sincerely,

Ruilin Lai, PhD

Central South University, China

Round 2

Reviewer 2 Report

Dear Authors, 

Thank you very much for professional explanations and taking my suggestions into account in the manuscript. I am deeply convinced that the article should be published and that it will be an important source of information for readers.
